# The Global Influence of Sodium on Cyanobacteria in Resuscitation from Nitrogen Starvation

**DOI:** 10.3390/biology12020159

**Published:** 2023-01-19

**Authors:** Markus Burkhardt, Johanna Rapp, Claudia Menzel, Hannes Link, Karl Forchhammer

**Affiliations:** 1Interfaculty Institute of Microbiology and Infection Medicine, University of Tübingen, Auf der Morgenstelle 28, 72076 Tübingen, Germany; 2CMFI, Bacterial Metabolomics, University of Tübingen, Auf der Morgenstelle 24, 72076 Tübingen, Germany

**Keywords:** cyanobacteria, dormancy, resuscitation, sodium, chlorosis

## Abstract

**Simple Summary:**

Dormancy and resuscitation are key processes to bacterial survival. In the absence of combined nitrogen sources, the non-diazotrophic model cyanobacterium *Synechocystis* sp. PCC 6803 turns into a metabolically quiescent state during a process termed chlorosis, enabling long-term survival. When nitrogen sources reappear, the cells resuscitate in a process that follows a highly orchestrated program. Here, we describe the essential role of sodium in the resuscitation process. We show that in addition to its role in the bioenergetics of chlorotic cells, sodium is involved in nitrogen compound assimilation, pH regulation, and the synthesis of key metabolites.

**Abstract:**

Dormancy and resuscitation are key to bacterial survival under fluctuating environmental conditions. In the absence of combined nitrogen sources, the non-diazotrophic model cyanobacterium *Synechocystis* sp. PCC 6803 enters into a metabolically quiescent state during a process termed chlorosis. This state enables the cells to survive until nitrogen sources reappear, whereupon the cells resuscitate in a process that follows a highly orchestrated program. This coincides with a metabolic switch into a heterotrophic-like mode where glycogen catabolism provides the cells with reductant and carbon skeletons for the anabolic reactions that serve to re-establish a photosynthetically active cell. Here we show that the entire resuscitation process requires the presence of sodium, a ubiquitous cation that has a broad impact on bacterial physiology. The requirement for sodium in resuscitating cells persists even at elevated CO_2_ levels, a condition that, by contrast, relieves the requirement for sodium ions in vegetative cells. Using a multi-pronged approach, including the first metabolome analysis of *Synechocystis* cells resuscitating from chlorosis, we reveal the involvement of sodium at multiple levels. Not only does sodium play a role in the bioenergetics of chlorotic cells, as previously shown, but it is also involved in nitrogen compound assimilation, pH regulation, and synthesis of key metabolites.

## 1. Introduction

Dormancy is one of the most widespread survival strategies in life [1]. In general, dormancy serves to endure unfavourable conditions. Those can range from short-term, for example, lack of sun at night, over long-term depletion of resources to endurance of harmful conditions. Bacteria commonly switch into a quiescent state when faced with nutrient limitation. This state of dormancy can range from long-lived, durable spores as found in, for example, *Bacillus subtilis* [2], over spore-like akinetes of some filamentous cyanobacteria [3] to the sole downregulation and reorganization of the cell metabolism, as observed, for example, in nutrient-deprived *Synechocystis* sp. PCC 6803 (hereafter *Synechocystis*) [4,5].

Cyanobacteria are one of the most primordial and ubiquitous groups of bacteria, and this is reflected in their morphological diversity and metabolic versatility, which enables them to adapt to different environmental conditions. One of the most limiting macronutrients is nitrogen [6,7]. *Synechocystis* is a unicellular, non-diazotrophic freshwater cyanobacterium. It is non-diazotrophic, therefore it is dependent on combined nitrogen sources such as nitrate or ammonium. When no nitrogen source is available, *Synechocystis* turns into a dormant state in a strictly regulated program termed chlorosis [5,8]. Cell growth arrests, the photosynthetic machinery is almost completely degraded, and after initial synthesis of the storage polymer glycogen, metabolic activities are highly reduced [4,5,9].

When a new nitrogen source is available, a genetically encoded and hierarchical resuscitation program begins. The immediate response is a major increase in ATP to revive cell anabolism [10]. During the first day, energy is produced heterotrophically through the degradation and respiration of glycogen. With the catabolism of glycogen, the cell receives reducing equivalents and carbon compounds for anabolic pathways, such as 2-oxoglutarate (2-OG) for the glutamine-synthetase-glutamate-synthase (GS-GOGAT) cycle [10]. The first genes that are expressed during resuscitation encode components of the translational machinery, RNA polymerases, and central metabolic reactions [9,11]. Subsequently, other cellular processes are activated, especially all components of the photosynthetic apparatus, involving a coordinated expression of the tetrapyrrole biosynthesis genes [11,12]. This leads to an intermediate, mixotrophic phase when the glycogen stores are still being degraded while photosynthesis and carbon fixation start again. When tracking the oxygen evolution, this phase is characterised by a gradual increase in light-dependent oxygen evolution, typically occurring between 20 and 30 h after the addition of a nitrogen source. After approximately 48 h, the glycogen storages are consumed, the cell metabolism is reconstituted, and the switch to vegetative growth occurs with the first cell division [11].

During the resuscitation period not all the assimilated nitrogen is used for protein synthesis. A part of the newly assimilated nitrogen is immediately secured in the nitrogen storage compound cyanophycin (CP) [13]. CP is a nitrogen storage polymer made of aspartate and arginine in equimolar amounts. The backbone is composed of poly-L-aspartic acid, with each carboxy side cahin linked to an arginine residue by an isopeptide bond. It is synthesised by the CP synthase CphA, a large, non-ribosomal peptide synthetase [14]. Typically in non-diazotrophic cyanobacteria, CP is synthesised during unbalanced nutritional conditions that limit growth, such as phosphate starvation [15,16]. In resuscitation however, the surplus of assimilated nitrogen, exceeding the anabolic demand, is temporarily stored as CP and may be mobilised during fluctuating nitrogen supply [13].

We previously showed that chlorotic cells have a specific requirement for sodium that differs from the sodium requirement of vegetative cells [17]. Vegetatively growing *Synechocystis* cells require sodium ions (Na^+^) primarily for the uptake of inorganic carbon. This can be explained mainly by the requirement for sodium to fuel two major bicarbonate uptake systems, SbtA and BicA [18]. Therefore, in the presence of elevated CO_2_ concentration, vegetative cells can grow in the absence of sodium. By contrast, during nitrogen starvation-induced chlorosis, the maintenance of a sodium motive force is required for membrane bioenergetics and ATP synthesis [17]. When chlorotic cells were supplemented with a combined nitrogen source, a rapid increase in ATP levels could be detected within a few minutes, which is the very first response in the resuscitation process. When resuscitation was started by the addition of NaNO_3_, the increase in ATP levels was higher than if resuscitation was started by the addition of KNO_3_. The surplus of ATP could be attributed to an increase in the sodium motive force by the addition of the sodium salt of nitrate. The same increase in ATP levels could also be obtained by just adding the same concentration of NaCl. However, how sodium affects the entire resuscitation program of *Synechocystis* was not clarified. To gain deeper insights into the awakening process of chlorotic *Synechocystis* and the role of sodium herein, we systematically analysed the requirement for sodium salts in the regreening of nitrogen-starved, chlorotic *Synechocystis* cells.

## 2. Materials and Methods

### 2.1. Cultivation and Growth Curves

Growth curves were generated in a Multicultivator OD-1000 with a gas mixing system GMS 150 (Photosystems Instruments, Dasov, Czech Republic). Vegetative cells were grown in a BG_11_ medium [19] in ambient air. Nitrogen starvation was induced by washing cells with the BG_11-0_ medium, which contains all BG_11_ components except for NaNO_3_. Cells were starved of nitrogen for at least 14 days at ~ 70 µE white light. To induce resuscitation, either just 17.5 mM KNO_3_ or additionally 17.5 mM NaCl was added to the culture. When comparing recovery with different N-sources, cultures were recovered with 10 mM KNO_3_ with or without 10 mM NaCl in comparison to cultures recovering with 10 mM NH_4_Cl with or without 10 mM NaCl. Resuscitation was performed in the presence or absence of 2% CO_2_ supplementation.

Recovery in ambient air was done in shaking flasks at 28 °C and approximately 70 µE white light. Samples were taken each day and measured in a photometer Heλios δ (Thermo Fisher Scientific, Waltham, MA, USA) at OD_750_. The whole cell spectrum was measured in a spectrophotometer Specord 50 (Analytik Jena GmbH, Jena, Germany).

### 2.2. Glycogen Measurement

Glycogen amounts were measured using an enzymatic assay according to [20], with modifications established by [21]. Two ml samples were taken, washed with water, and incubated in 30% KOH for 2 h at 95 °C. Then, ice-cold ethanol was added to a final concentration of 75%, and glycogen precipitated overnight at −20 °C. Samples were then washed with 70 and 98% ethanol, spun down, the pellet dried, and the glycogen digested by the addition of a solution of 1 00 mM sodium acetate and 4.4 U/µL amyloglucosidase at pH 4.5 for 2 h. Then, 200 µL of the samples were mixed with 1 mL of 6% O-toluidine in acetic acid and incubated for 10 min at 100 °C. Absorbance was then measured at 635 nm using a Tecan Spark 10M (Tecan, Männerdorf, Switzerland). A glucose calibration curve was used to determine the glycogen amount in the samples. At least three biological replicates were measured for every condition.

### 2.3. ATP Determination in the Cells

One ml samples were taken from each culture in cultivation conditions and immediately frozen in liquid nitrogen. Samples were then stored at −80 °C until further processing. Cells were lysed by three cycles of cooking at 99 °C and flash freezing in liquid nitrogen. Debris was spun down at 25,000 g at 4 °C for 1 min. The ATP content in the supernatant was measured according to the instructions of the “ATP determination kit’’ (Molecular Probes (A22066), Eugene, OR, USA). A 50 µL reaction mix containing reaction buffer, luciferin, and firefly luciferase was mixed with a 10 µL sample supernatant and measured in a Sirius Luminometer (Berthold Detection Systems, Bad Wildbad, Germany). An ATP calibration curve was used to determine the ATP amount in the samples. At least three biological replicates were measured for every condition.

### 2.4. PS2 Yield with Pulse Amplitude Modulation

The yield of Photosystem II (PSII) was measured in vivo using a WATER-PAM chlorophyll fluorometer (Walz GmbH, Effeltrich, Germany). The maximum PSII quantum yield was determined by saturation pulse. At least three biological replicates were measured, and each one in three technical replicates.

### 2.5. Nitrate/Nitrite Measurement in the Growth Medium

One mililiter of sample was harvested and centrifuged at 13,000× *g* for 5 min. Nitrate and nitrite were quantified by measuring the absorbance at 210 nm in the cell-free medium. The nitrate values were corrected for the presence of nitrate [22]. To measure, nitrite 300 µL of the cell-free sample was added to 300 µL sulfanilamide solution (1% sulfanilamide in 5% phosphoric acid) and incubated for 10 min at room temperature in the dark. Afterwards, 300 µL NED solution (0.1% N-1-napthylethylenediamin dihydrochloride in water) was added and incubated in the dark for 10 min. Then, the absorbance at 530 nm was measured [23].

### 2.6. Ammonium Measurement in the Growth Medium

A dilution series from 0 to 2 mM NH_4_Cl was prepared. Samples of 1 mL of each culture were taken and spun down; the supernatant was moved to a new reaction tube. A nessler reagent of 20 µL (containing K_2_[HgI_4_]) was mixed with 980 µL of the supernatant or the dilution series. The mixture was transferred to a cuvette, and the absorbance was measured at 410 nm [24].

### 2.7. Metabolome Measurement

Cells were grown to an OD_750_ > 0.4. To sample, 10 mL were filtered through pore-size 1.2 µM filters (WHA1822025, cytiva, Marlborough, MA, USA). Filters were put into reaction tubes, frozen in liquid nitrogen, and stored at –80 °C until further use. The filters were thawed in 500 µL acetonitrile:methanol:water (40:40:20) at −20 °C. Filters were incubated in the extraction solvent for 4 h at −20 °C. Metabolites were extracted from the filter by pipetting the solvent up and down, and the supernatant was then moved to a new tube. To ensure cell lysis glass beads were added to the supernatant and ribolysed at 6.5 m/s for 30 s in 2 cycles and a 5 min break in between, centrifuged at >13,000× *g* for 15 min at −9 °C, and the supernatant transferred to a new tube once more. The samples were then stored at −80 °C until further use. The measurement was done as described in [25]. Metabolomic analysis was performed via LC-MS/MS (Agilent TQ6495, Santa Clara, CA, USA). Relative quantification was performed by adding a 13C internal standard.

### 2.8. Oxygen Evolution Measurement

Oxygen evolution was measured using a Clark-type oxygen electrode DW1 (Hansatech, King’s Lynn, Norfolk, UK) as described in [17].

### 2.9. PH Measurement

Extracellular pH was measured using a pH electrode (InLab Micro, Mettler-Toledo, Columbus, OH, USA). At least three biological samples were measured during each sampling point.

Intracellular pH was measured using the fluorescent indicator BCECF (2′, 7′-bis-(2-carboxyethyl)-5-(and-6)-carboxyfluorescein) (Invitrogen AG, Waltham, MA, USA). Samples were taken and mixed with BCECF for a final concentration of 0.5 µM and incubated in the dark for 15 min. Measurement was done in a Spark 10M (Tecan, Männerdorf, Switzerland) with BCECF emission at 535 nm and excitation at either 490 nm or 439 nm. Additionally, measurements at the most acidic point and the most alkaline point of the linear phase were required. To detect those, a calibration curve was made by washing cells free of the medium and resuspending them in BG0 10 mM Hepes with a pH ranging from 4 to 10, in 1 pH increments. BCECF was added to a final concentration of 0.5 µM and samples were incubated in the dark for 5 min before adding a CTAB solution to a final concentration of 0.4% and incubated another 10 min in the dark before measurement in the Spark 10M. The ratio between excitation at 490 nm and 439 nm was calculated, and the necessary measurement points are chosen. The intracellular pH was then calculated using the formula:(1)$$pH=pK_{a}-log\frac{\left(R-R_{A}\right)}{\left(R_{B}-R\right)}*\frac{F_{A\left(439\;\mathrm{nm}\right)}}{F_{B\left(439\;\mathrm{nm}\right)}}$$

With *pK_a_* being the pK of the medium, *R* being the ratio of signal between excitation at 490 nm and 439 nm, and *F* the fluorescent signal. *A* indicates the most acidic point of the linear range, and *B* is the most basic.

### 2.10. Sakaguchi Staining and Bright Field Microscopy

CP granules were visualised by using the arginine-selective Sakaguchi staining method, according to [26].

Photographs were taken with a Leica DM2500 microscope, and a Leica DFC420C color camera and Leica Application Suite Software. Microscope slides were covered in dried 1% (*w/v*) agarose solution to immobilise the cells.

### 2.11. Transmission Electron Microscopy

Samples for TEM were prepared, and images were taken as described by [26].

## 3. Results and Discussion

### 3.1. Resuscitation in Absence of Sodium

To investigate the requirement for sodium in the resuscitation process of chlorotic *Synechocystis*, cultures that had been starved of nitrogen for 2 weeks were used. Before starting resuscitation, the cells were washed and resuspended in a sodium-free BG11 medium. Immediately thereafter, either 10 mM KNO_3_ or 10 mM NH_4_Cl was added, respectively, with or without the addition of 10 mM NaCl. Then, resuscitation was studied for the following 2 to 3 days. The same type of experiment was carried out either at ambient air in flasks or in tubes that were bubbled with air enriched with 2% CO_2_. Cells that were incubated in ambient air were unable to regreen and recover in the absence of Na^+^ (Figure 1A, top). The lack of reconstitution of photosynthetic pigments is also visible in the spectrum from 600 to 750 nm after 2 days of resuscitation, where in recovered cells, phycocyanin absorbs light at 630 nm and chlorophyll α at 680 nm (Figure 1B). A further indication of a reconstitution of the photosynthetic activity can be obtained by measuring the quantum yield of photosystem (PS) 2 by pulse amplitude modulation (PAM fluorometry and saturation pulse method). During resuscitation at standard conditions, the PAM signal first drops to 0, followed by an increase starting after 20 h [11]. Here, the PAM signal of the cultures in the presence of sodium increased within 24 h when recovered with nitrate and 48 h when recovered with ammonium. By contrast, in the absence of Na^+^, no increase in the PAM signal was detectable (Figure 1C, top). This clearly indicated that in the absence of Na^+^, the cells were unable to restore the photosynthetic machinery. Energy production during early resuscitation is solely based on the degradation and respiration of glycogen, and as soon as photosynthetic activity appears, oxygen evolution increases gradually [11]. When resuscitation was performed in the absence of sodium, cells had a higher respiration rate and were unable to turn on oxygen evolution (Figure 1D), another indication of the lack of photosynthetic activity. However, the degradation of glycogen was unaffected by the presence or absence of sodium (Figure 1E). The last step of resuscitation is the initiation of cell division indicated by an increase in OD_750_. This increase was not observable when resuscitation was performed in the absence of sodium, while the OD_750_ started to increase in presence of Na^+^ after about 48 h of resuscitation (Figure 1F). Since these first experiments clearly indicated that resuscitation required Na^+^ ions, we asked whether we could bypass the requirement for Na^+^ by providing the resuscitating cells with 2% CO_2_, as observed for vegetative cells [17]. However, the presence of CO_2_ did not enable cells to resuscitate successfully, although a slight green colour appeared after 2 days of incubation (Figure 1A, bottom). The PAM signal indicated a slight recovery of PS2 quantum yield towards 48 h of resuscitation (Figure 1C, bottom), although this minute signal PSII activity was not sufficient to enable oxygen evolution (Figure 1D, bottom). The cells remained in this incomplete recovered state after prolonged incubation, indicating that the partial regreening and traces of PSII activity did not support the resuscitation process. Glycogen consumption was comparable to that in ambient air (Figure 1E), indicating that glycogen catabolism and respiration are not dependent on Na^+^ supply. The failure of the cells to switch back to vegetative growth in the absence of sodium is also evidenced by the stagnation of OD_720_ (Figure 1F, bottom). All these results indicate that in contrast to vegetative cells, which require sodium to import carbon, resuscitating cells require sodium for functions other than carbon acquisition.

We asked ourselves how much sodium is necessary for *Synechocystis* to be able to survive and potentially grow in ambient air. We thus cultivated cells in increasing concentrations of NaCl and discovered that in continuous light, cells require around 100 µM of NaCl in the medium to survive, while in a 12 h day/12 h night cycle, they needed at least 500 µM (Appendix A).

### 3.2. Assimilation of Nitrogen Sources and pH

Based on these results, we next investigated the role of sodium in the acquisition and assimilation of the nitrogen sources during resuscitation from nitrogen chlorosis. Here, we used potassium nitrate or ammonium chloride as nitrogen sources. Since nitrogen is assimilated in the form of ammonium, nitrate must be reduced to nitrite and subsequently to ammonia (Figure 2A). Then, the glutamine synthase (GS) catalyses the incorporation of ammonia into glutamate forming glutamine. Ammonium as an N-source is co-transported across the membrane as ammonia together with a proton and can be immediately incorporated into the GS-GOGAT cycle (Figure 2A). When cells were resuscitated with 1 mM ammonium, its concentration in the medium decreased both in the presence or absence of sodium. However, the levels decreased faster in presence of Na^+^, reaching residual levels after 24 h of resuscitation, while in absence of sodium, this took 48 h (Figure 2B). To measure the consumption of nitrate, cells were recovered with 1 mM KNO_3_ either in the presence or absence of sodium, and the extracellular concentrations of nitrate and nitrite were measured over a time period of 48 h. The concentration of nitrate stayed largely unchanged in the first 12 h. Thereafter, in the presence of sodium, extracellular nitrate decreased dramatically and was almost completely consumed after 48 h. By contrast, no significant consumption of nitrate was detected in the absence of sodium (Figure 2C). Nitrite is an intermediate in nitrate reduction and is excreted when nitrate reduction exceeds nitrite reduction [22]. During resuscitation in the presence of sodium, the nitrite levels first increased until 24 h and then dropped again at 48 h, as the cells were nitrogen depleted again. In the absence of Na^+^, nitrite levels initially increased even faster than in the presence of sodium until 12 h and then stayed constant until 48 h (Figure 2D). In agreement with the lack of nitrate consumption, no excretion of ammonium could be detected in the sodium-free culture, either (Appendix A). This clearly indicates that the initial uptake of nitrate and nitrate reductase activity is not impaired in the absence of sodium, whereas the reduction to ammonium seems to be impaired.

The reduction of nitrate to ammonia requires nine protons in total. Given the limited number of available protons in a *Synechocystis* cell [27,28,29], supplying *Synechocystis* with nitrate as the sole N-source should lead to a cytoplasmic alkalisation unless sufficient protons can be imported. By contrast, when the cells are using ammonium as a nitrogen source, they take up ammonia in co-transport with a proton [30], which delivers new protons into the cytoplasm and should thereby cause its acidification. Therefore, we next measured the extra- and intracellular pH changes during resuscitation to reveal how the presence of sodium affects pH homeostasis. When resuscitation was performed with nitrate in the presence of Na^+^, the extracellular pH increased in the second phase of resuscitation (after 20 h). As expected, the extracellular pH remained largely unchanged in the absence of sodium, as no nitrate was consumed (Figure 2E). By contrast, the measurement of intracellular pH using the trapped fluorescent indicator method with BCECF yielded intriguing results. In the presence of sodium, the intracellular pH transiently increased from pH 7.5 to approximately 7.9, returning to initial levels after 6 h and then remaining constant. By contrast, in the absence of sodium, an immediate increase in intracellular pH occurred which further increased during incubation, reaching a value of 8.3 (Figure 2F). This indicates a lack of intracellular protons, which are generally imported by sodium-dependent antiporters like the Na^+^/H^+^ antiporter of the NhaS-family [31].

When resuscitation was triggered by the addition of ammonium, transient acidification was measured both in the presence or absence of ammonium (Figure 2F). Likewise, no difference in extracellular pH was observed (Figure 2E). When fed with ammonium, cells appear to regulate the pH independently of the availability of sodium. Nevertheless, also under these conditions, resuscitation in the absence of sodium failed, indicating the existence of further sodium-dependent processes.

### 3.3. Metabolome of Resuscitating Cells

To gain deeper insights into the metabolic changes during resuscitation with nitrate in the presence or absence of sodium, we investigated the metabolome during the first 12 h of resuscitation. In this early phase of resuscitation, no CO_2_ fixation takes place, and all cellular carbon is derived from the catabolism of glycogen stores [10]. Therefore, during this heterotrophic phase, sodium should be required for cellular processes other than carbon acquisition. Metabolites were extracted using glass beads to mechanically disrupt the cells in a solvent made of acetonitrile, methanol, and water, and their identity and relative concentration were determined using MS.

The method employed allowed the determination of changes in amino acids and a few central carbon metabolites. This is the first global description of the amino acid steady-state level in the resuscitation of *Synechocystis*, to our knowledge.

The amino acid steady-state levels are the result of amino acid synthesis and its consumption by downstream metabolic pathways and protein synthesis. Overall, for many amino acids, the steady-state level only subtly changed in response to the presence or absence of sodium, although in the absence of sodium, the assimilation of nitrogen was clearly abrogated. This indicates the presence of efficient regulatory mechanisms to keep the steady-state levels of amino acids constant. Therefore, we will focus the analysis of the experiment on examples showing deviations from the standard condition in the absence of sodium.

In the presence of Na^+^, glutamate was the amino acid whose cellular concentration increased the most within the first hour of resuscitation (Figure 3A). Thereafter, a further steady increase up to 10 fold was observed after 12 h as compared to the chlorotic cells. This is reasonable since glutamate is the net product of nitrogen assimilation through the GS-GOGAT cycle. In the absence of Na^+^, the levels of glutamate also slightly increased, but the steady-state level increased only to about half the concentration. In contrast to glutamate, the concentration of glutamine remained quite constant throughout early resuscitation in the presence of sodium, indicating a thoroughly balanced activity of glutamine synthesis by GS and its consumption by GOGAT. However, in the absence of sodium, glutamine levels dropped and stayed lower, indicating that GS activity lagged behind GOGAT. The imbalanced GS/GOGAT cycle in the absence of sodium could have several reasons. First, the inability to reduce nitrate to ammonia could result in substrate limitation for the GS reaction. Second, the inability to control the intracellular pH could reduce GS enzyme activity. Third, in the absence of sodium, energy metabolism is impaired [17], thus impairing the ATP-dependent GS reaction.

A similar pattern as for glutamine was observed for arginine. Since arginine is derived from glutamate and carbamoyl phosphate, whose synthesis is glutamine derived, the reduced levels of arginine in the absence of sodium could be explained by decreased nitrogen assimilation. A strikingly different pattern was observed for proline, the last member of the oxoglutarate/glutamate-derived amino acid family. Proline levels stayed constant during resuscitation in the presence of sodium, while the levels continuously increased in the absence of Na^+^. Proline can be derived from the ArgZ/ArgE catalysed arginine catabolism, which is part of the nitrogen-assimilatory ornithine-ammonia cycle [32]. The N-terminal part of ArgZ displays arginine dihydrolase activity, transforming arginine into ornithine, CO_2_, and ammonia. Ornithine can then re-enter arginine synthesis or can be converted by ornithine cyclodeaminase activity into proline, presumably catalysed by the C-terminal domain of ArgZ [33]. The increasing proline levels in the absence of Na^+^ indicate that ornithine cyclodeaminase activity exceeds the re-entry of ornithine in the arginine pathway due to impaired nitrogen assimilation. Furthermore, proline is known to function as an osmoprotectant in many bacteria [34,35,36], and therefore, the increasing proline levels could enhance the protection against osmotic stress.

In the family of oxaloacetate/aspartate amino acids, only subtle differences were observed (Figure 3B). The lower levels of asparagine in the absence of sodium reflect the lower levels of glutamine, which is required for asparagine synthesis.

In the family of aromatic amino acids, tyrosine showed a remarkable pattern. Its levels decreased 2 fold during resuscitation in the presence of sodium, whereas the levels even slightly increased in the absence of sodium (Figure 3C). Increased levels of tyrosine in N-starved *Synechocystis* are in agreement with a recent study reporting a strong accumulation of tyrosine levels when *Synechocystis* cells are subjected to nitrogen deprivation [37]. The levels of phenylalanine also increased during resuscitation in absence of sodium (Figure 3C). No function has been ascribed to free phenylalanine or tyrosine to our knowledge, except in protein synthesis. Thus, we can assume that the drop in tyrosine levels occurring during successful resuscitation reflects its consumption by protein synthesis. The elevated levels in the absence of sodium would then be caused by impaired protein synthesis.

Amino acids from the pyruvate family generally showed lower levels in the absence of Na^+^(Figure 3D). For all three, alanine, valine, and isoleucine, a significant increase from 6 to 12 h in resuscitation with sodium was noticeable. This is likely due to the successful re-establishment of the core anabolic reactions, increasing the pool sizes of these amino acids for protein synthesis and further metabolic reactions.

The measured intermediates of the TCA cycle also showed interesting behaviour. The levels of citrate and malate were comparable between the two conditions whereas a large difference was visible for succinate, with a 350% increase at 12 h of resuscitation in the absence of sodium (Figure 3E). Succinate is produced in the modified TCA cycle reactions of cyanobacteria from 2-OG by 2-Oxoglutarate decarboxylase and succinic-semialdehyde dehydrogenase [38]. Since 2-OG can no longer be efficiently converted to glutamate due to the impaired GS-GOGAT cycle in the absence of sodium, the synthesis of succinate is apparently increased under these conditions.

### 3.4. Consequential Effects on Cyanophyicn and the Cytoplasm

Cyanophycin (CP) is a nitrogen storage polymer composed of an aspartate backbone and arginine sidechains. The synthesis of this polymer is an indicator of global nitrogen availability to non-growing cells, as an excess of assimilated nitrogen, not needed for protein synthesis, is stored in this nitrogen-rich polymer. When chlorotic *Synechocystis* cells are resuscitated by nitrate addition, they transiently produce CP during the first day of resuscitation as a nitrogen reservoir to cope with fluctuations in nitrogen supply [13]. To reveal whether this process also depends on Na^+^, we assessed CP production using the Sakaguchi reaction, which stains arginine residues. CP granules in cells are visible as red dots [13]. Here, we focused on cultures before and after 12 h of resuscitation. As shown in Figure 4, cells that were resuscitated with nitrate but in the absence of sodium were strongly impaired in CP synthesis, while in the presence of sodium, the expected synthesis of CP could be observed. When cells were resuscitated in the presence of ammonium, a small amount of CP synthesis could be observed.

Since the absence of Na^+^ has global consequences for the physiology of the cells, we also investigated cell morphology using transmission electron microscopy. We took samples before the addition of KNO_3_ and 24, 48, and 66 h after. During resuscitation in standard conditions, between time points 0 and 24 h +N, the glycogen storages should largely disappear, and ribosomes, CP granules, and an increase in thylakoid membranes should be noticeable. Thereafter, CP granules should disappear again, whereas thylakoid membranes and carboxysomes should appear [11]. These morphological changes were visible in the cultures recovered under standard conditions (Figure 5B). By contrast, the cells that were incubated in the absence of Na^+^ appeared to lose intracellular structures (Figure 5A). This indicates that the longer the recovering cells are left without sodium, the more they degrade any intracellular structures to the point of looking like ghost cells.

## 4. Conclusions

We have shown in this study that chlorotic cells of *Synechocystis* have a multi-levelled requirement for sodium during resuscitation from nitrogen depletion. Sodium has been previously reported to have a significant role in bioenergetics during the exit from the chlorotic state and in carbon import during vegetative growth [17]. Here we show that throughout resuscitation from nitrogen depletion-induced chlorosis, sodium is required from multiple processes. What are the differences in sodium requirement between vegetatively growing cells and cells resuscitating from chlorosis?

In vegetative cells, the primary requirement for sodium is to enable bicarbonate uptake through the sodium dependence of the major bicarbonate uptake systems SbtA and BicA [18,39,40]. Therefore, at elevated CO_2_ levels, which circumvents high-affinity bicarbonate transport, vegetative growth can take place in the absence of sodium. In contrast, in resuscitating cells, supplementation of cells with 2% CO_2_, despite enabling synthesis of small amounts of chlorophyll, does not restore full recovery of cells, indicating further requirements for sodium beyond the inorganic carbon supply.

When resuscitation is started by the addition of nitrate, sodium rather helps in the acquisition thereof and the control of intracellular pH. This is because to reduce nitrate to ammonia, cells require nine protons, which requires efficient pH control in the cytoplasm. Na^+^/H^+^ antiporters can supply the cytoplasm with protons. In the absence of sodium, proton import, and thus, pH control, is impaired leading to the alkalisation of the cytoplasm. Nevertheless, the nitrate reductase reaction, which requires 2 protons, still takes place, and the produced nitrite is now excreted to the medium instead of being further reduced to ammonium. By contrast to nitrate, when fed with ammonia in the absence of sodium, cells are able to take up ammonium but still are unable to incorporate it into proteins or storage compounds like CP. This indicates that central reactions of nitrogen assimilation are also impaired by the absence of sodium.

Metabolite analysis indicated an imbalanced GS/GOGAT cycle in the absence of sodium (see above). The GS reaction appears to be most strongly affected by the lack of sodium, as evidenced by decreasing glutamine levels in sodium-free cells. In addition to the impaired supply of substrate through nitrate reduction, it is likely that the high energy demand of GS cannot be satisfied without the use of sodium bioenergetics, which provides the basis for efficient ATP synthesis in chlorotic cells [17]. This imbalance in one of the most essential metabolic cycles of *Synechocystis* will lead to further problems downstream. The increased levels of proline during resuscitation in absence of sodium, for example, could be derived from arginine catabolism by ArgZ [33]. Supporting this, the expression of *argZ* and the protein levels of ArgZ also increase in early resuscitation [11,12]. Proline would then serve as an osmolyte to stabilise the cytoplasm while the glycogen stores and the intracellular structures are degraded, as witnessed in the TEM analysis. The cells recovering in the absence of sodium degrade their cytoplasmic structures until they appear empty. This is reminiscent of the degradation and recycling processes occurring in lysosomes during the autophagy of eucaryotic cells [41]. It is unclear whether the increased level of tyrosine in chlorotic cells simply reflects the lack of catabolism of free tyrosine during chlorosis or whether it is of functional significance. In specific enzymes, tyrosine residues may act as radical quenchers [42]. Whether free tyrosine could also function as a radical quencher has, to our knowledge, never been demonstrated, but if so, it would provide additional protection to chlorotic cells.

All these results highlight the tight interconnection of cellular processes through the pivotal role of ions such as sodium, protons, and small metabolites. The broad effects of ion homeostasis could be revealed by studying cells in an extreme metabolic situation, with limited possibilities to compensate for external perturbations. In a broader scope, such studies deepen our understanding of the physiology of dormant cells and the reawakening after quiescence.

## Figures and Tables

**Figure 1 biology-12-00159-f001:**
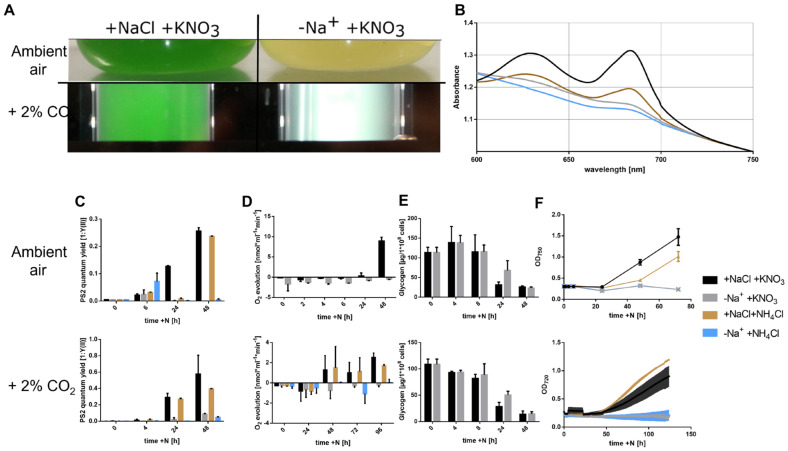
Resuscitation in the absence of Na^+^ is impossible. (**A**) shows two cultures after 14 days of chlorosis and two days of resuscitation in ambient air (top) or supplemented with 2% CO_2_ (**bottom**), the left in presence of Na^+^ and the right in absence thereof. (**B**) shows the absorbance of recovering cultures in ambient air from 600 nm to 750 nm. (**C**) shows the quantum yield of PSII as a measure of photosynthetic activity. (**D**) shows the oxygen evolution during resuscitation. (**E**) shows glycogen consumption. (**F**) shows the OD at 750 nm (top) or 720 nm (bottom) as an indicator of cell mass. (**C**–**E**) top graphs refer to cultures in ambient air, bottom graphs refer to cultures in a 2% CO_2_ environment. Samples were taken 0, 4, 6, 24, and 48 h after the addition of 10 mM KNO_3_ or 10 mM NH_4_Cl, respectively. Each data-point represents measured triplicates. Error bars represent the SD.

**Figure 2 biology-12-00159-f002:**
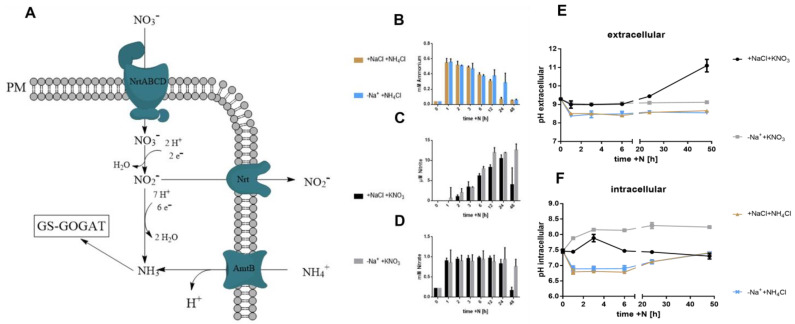
Nitrogen import and incorporation. Time-course analysis of nitrogen compounds and pH during resuscitation. Resuscitation was initiated by the addition of KNO_3_ or NH_4_Cl to a final concentration of 10 mM. (**A**) depicts a model of the incorporation of combined nitrogen sources into the cell metabolism. GS = Glutamine synthase, GOGAT = Glutamate Oxoglutarate Amidtransferase, PM = Plasmamembrane. Ammonium (**B**) and nitrate (**C**) levels are shown in mM/10^8^cells, nitrite levels (**D**) in µM/10^8^cells. (**E**) shows the extracellular pH, (**F**) the intracellular. Samples were taken 0, 1, 2, 3, 6, 12, 24, and 48 h after the addition of a combined nitrogen source (KNO_3_ or NH_4_Cl). Bars represent measurement of biological triplicates. Error bars represent the SD.

**Figure 3 biology-12-00159-f003:**
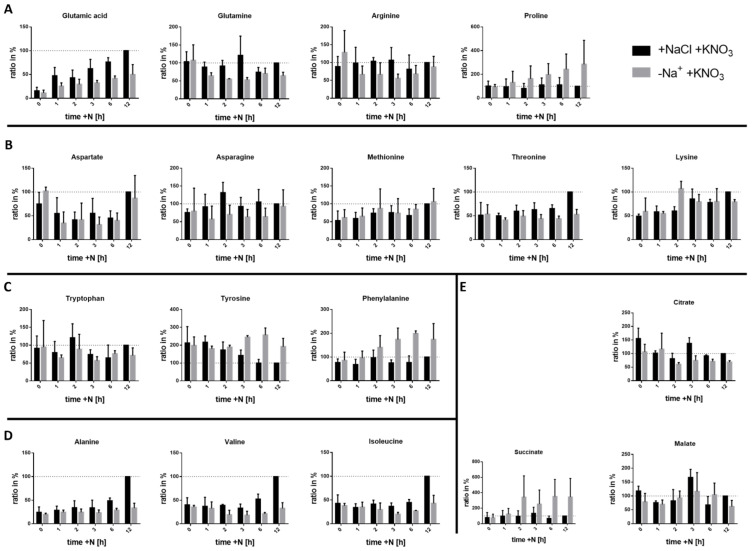
Levels of metabolites during resuscitation in the presence or absence of Na^+^. Cells were recovered with 17.5 mM KNO_3_ after 2 weeks of nitrogen starvation. (**A**) shows the 2-OG amino acid family, (**B**) the oxaloacetate family, (**C**) the aromatic amino acids and (**D**) the pyruvate family. (**E**) shows intermediates of the TCA cycle. The *y*-axis depicts the ratio in % normalised to 12 h of resuscitation in the presence of sodium, the *x*-axis shows the time after the addition of nitrate in hours. Bars represent measurement of biological triplicates. Error bars represent the SD.

**Figure 4 biology-12-00159-f004:**
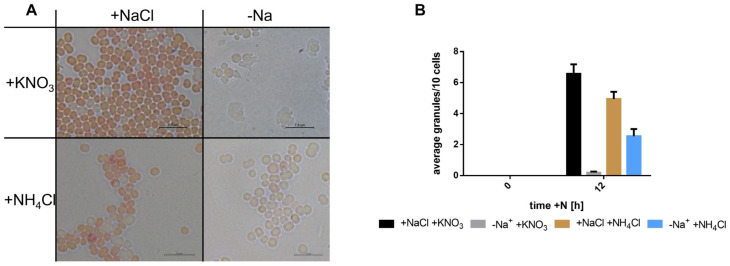
Cyanophycin production in the presence and absence of sodium. (**A**) shows pictures of cells after 12 h of resuscitation with nitrate or ammonium, in the presence or absence of sodium. Cells were stained with the Sakaguchi staining method. Scale bar depicts 7.5 µm in each picture. (**B**) depicts the amount of cyanophycin granules per cell during resuscitation, counted after Sakaguchi staining. Samples were taken before start of resuscitation and 12 h after start. At least 200 cells were counted per sample per time point.

**Figure 5 biology-12-00159-f005:**
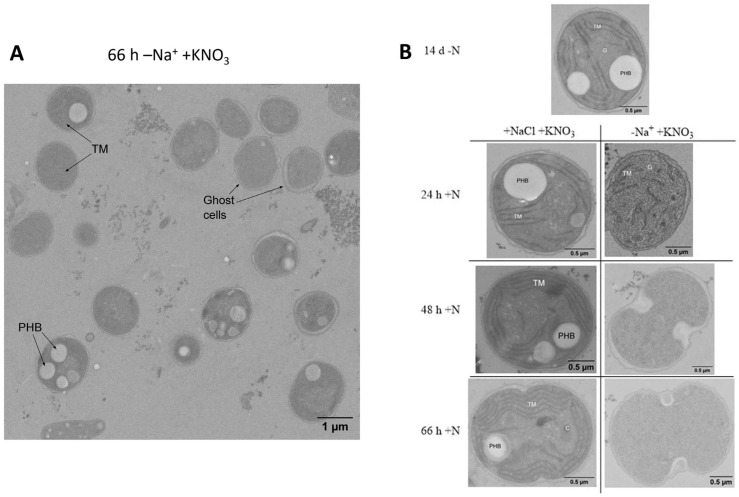
TEM pictures of resuscitating cells in the presence and absence of sodium. (**A**) provides an overview of multiple cells 66 h after the start of resuscitation in the absence of sodium. (**B**) depicts individual cells in resuscitation. On the left side are cells recovering in the presence of sodium, on the right in the absence. Samples were taken before resuscitation (after 14 days of chlorosis) and 24, 48, and 66 h after initiation. TM = thylakoid membrane, PHB = polyhydroxybutyrate, G = glycogen, C = carboxysome. Scale bar in (**A**) represents 1 µm, bars in (**B**) represent 0,5 µm.

## Data Availability

All data obtained during this work is available from the authors on request.

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
