# Peer review of "The Global Influence of Sodium on Cyanobacteria in Resuscitation from Nitrogen Starvation"

_biology, 2023, doi:10.3390/biology12020159_

Round 1
Reviewer 1 Report
The manuscript by Burkhardt and colleagues is devoted to resuscitation of Synechocystis 11 sp. PCC 6803 cells in response to addition of different nitrogen sources. It is shown that resuscitation essentially depends on the presence of sodium ions in the medium.
The manuscript can be published after a minor revision.
Major comments.
1. The usage of four “shades of gray” for different growth conditions in Fig. 1, 2, 4, and S2 is extremely confusing. These shades look alike on the bar charts. Therefore, it is hardly possible to identify the growth conditions. The simplest solution is to label the four growth conditions with four different colors throughout the manuscript.
2. Line 279 and below. The Fig. 2 is confusing as whole. The Panel A is in its middle, after the panels B,C, and D. In the text, the panel D is mentioned in the text before the panels B and C.
One possible solution is to move the panel A, as a separate figure, to the Introduction, which will allow increasing the size of other panels and using larger fonts for labeling. Panels B-F should be arranged in the same order in which they are mentioned in the text. Also, as mentioned above, four different colors should be used for different growth conditions.
3. Line 306. Better to delete the sentence on 251 protons in the cytoplasm. Most likely, this number is wrong; we cannot provide any numbers until we know exactly the inner pH-buffering capacity of the cell, and we don't know it.. The subsequent sentence that the operation of nitrogen fixating enzymes should lead to alkalinization of the cytoplasm is convincing and fully sufficient.
4. Line 412 and below: the succinate dehydrogenase (SDH) transfers no protons across the membrane. The high intracellular pH can only affect the membrane proton pumps that follow SDH in the electron transfer chain, such as the cytochrome b6f complex and cytochrome oxidase.
5. Line 466 and following: As already indicated, better to delete all the mentions of 251 proton.
Minor comments:
Line 58: better to use “reducing equivalents”
Line 61: what is “transitional machinery”? Perhaps, translation(al) machinery?
Line 73-74: it is advisable to describe in more details the structure of cyanophycin or to provide its structure somewhere. The current description is not quite informative.
Line 112: complete the sentence “70 µE white light”.
Line 127: complete the title “… determination in the cells”
Line 143: complete the title “… measurement in the growth medium”
Line 152: complete the title “… measurement in the growth medium”
Line 354. The sentence “Therefore, we will focus the analysis of the experiment to prominent examples” should be re-written in a more comprehensive way.
Author Response
Response to Reviewer 1 Comments
Major Comments:
Point 1:
The usage of four “shades of gray” for different growth conditions in Fig. 1, 2, 4, and S2 is extremely confusing. These shades look alike on the bar charts. Therefore, it is hardly possible to identify the growth conditions. The simplest solution is to label the four growth conditions with four different colors throughout the manuscript.
Response 1:
The figures were adapted so that the different conditions are coloured black (+NaCl +KNO3), grey (-Na+ +KNO3), brown (+NaCl +NH4Cl) and blue (-Na+ +NH4Cl).
Point 2:
Line 279 and below. The Fig. 2 is confusing as whole. The Panel A is in its middle, after the panels B,C, and D. In the text, the panel D is mentioned in the text before the panels B and C.
One possible solution is to move the panel A, as a separate figure, to the Introduction, which will allow increasing the size of other panels and using larger fonts for labeling. Panels B-F should be arranged in the same order in which they are mentioned in the text. Also, as mentioned above, four different colors should be used for different growth conditions.
Response 2:
Panel A was moved to the left. Original panel D was set to be new panel B, original panel B to be new panel C and original panel C to be new panel D. Mentions in the text were accordingly adapted.
Point 3:
Line 306. Better to delete the sentence on 251 protons in the cytoplasm. Most likely, this number is wrong; we cannot provide any numbers until we know exactly the inner pH-buffering capacity of the cell, and we don't know it. The subsequent sentence that the operation of nitrogen fixating enzymes should lead to alkalinization of the cytoplasm is convincing and fully sufficient.
Response 3:
Mention of specific amounts of free protons available has been deleted.
Point 4:
Line 412 and below: the succinate dehydrogenase (SDH) transfers no protons across the membrane. The high intracellular pH can only affect the membrane proton pumps that follow SDH in the electron transfer chain, such as the cytochrome b6f complex and cytochrome oxidase.
Response 4:
The text was changed to describe that the increased level of succinate is caused as downstream effect of missing incorporation of 2-OG into the GS/GOGAT cycle. Removed mention of further processing by succinate dehydrogenase.
Point 5:
Line 466 and following: As already indicated, better to delete all the mentions of 251 protons.
Response 5:
Mention of specific amounts of free protons available has been deleted.
Minor Comments:
Point 6: Line 58: better to use “reducing equivalents”
Response 6: Line 58: changed “reduction euqivalents” to “reducing equivalents”
Point 7: Line 61: what is “transitional machinery”? Perhaps, translation(al) machinery?
Response 7: Line 61: changed “transitional machinery” to “translational machinery”
Point 8: Line 73-74: it is advisable to describe in more details the structure of cyanophycin or to provide its structure somewhere. The current description is not quite informative.
Response 8: Line 73 – 74: described the distribution of aspartate and arginine and structure of cyanophycin more in detail
Point 9: Line 112: complete the sentence “70 µE white light”.
Response 9: Line 113: added “70 µE white light”
Point 10: Line 127: complete the title “… determination in the cells”
Response 10: Line 128: completed the title with “determination in the cells”
Point 11: Line 143: complete the title “… measurement in the growth medium”
Response 11: Line 144: completed the title with “measurement in the growth medium”
Point 12: Line 152: complete the title “… measurement in the growth medium”
Response 12: Line 153: completed the title with “measurement in the growth medium”
Point 13: Line 354. The sentence “Therefore, we will focus the analysis of the experiment to prominent examples” should be re-written in a more comprehensive way.
Response 13: Line 354 – 355: further defined “prominent examples”
Reviewer 2 Report
Major comments
1. The reason why requirement for sodium is different between resuscitating cells or chlorotic cells and vegetative cells should be explained from a standpoint of the physiological significance or at least be discussed in Discussion, if possible, although it is described in the text that in contrast to vegetative cells, which require sodium to import carbon, resuscitating cells require sodium for functions other than carbon acquisition.
Minor comments
1. The results measured at 24 and 48 hours after start are not given in Figure 4.
2. Letters showing G (glycogen) and C (carboxysome) are not written or invisible in Figure 5.
3. It seems to me that the result of TEM picture (Figure 5 (A)) shows a mixed state of resuscitating cells at (66h -Na + KNO3). On the contrary, one ghost cell can be seen in the TEM picture given in Figure 5 (B) (right side bottom) in spite of the cell under the same situation of (66h -Na + KNO3).
4. “Cyanophycn” in the subtitle “3.4. Consequential effects on cyanophyicn and the cytoplasm” should be “Cyanophycin”.
5. In conclusion, periods are doubly put after [36] in the phrase “during autophagy of eucaryotic cells [36]”. Therefore, one period should be removed.
Author Response
Response to Reviewer 2 Comments:
Major comments
Point 1: The reason why requirement for sodium is different between resuscitating cells or chlorotic cells and vegetative cells should be explained from a standpoint of the physiological significance or at least be discussed in Discussion, if possible, although it is described in the text that in contrast to vegetative cells, which require sodium to import carbon, resuscitating cells require sodium for functions other than carbon acquisition.
Response 1: Re-wrote the conclusions to explain the different requirements for sodium in vegetative growth versus chlorosis and resuscitation.
Minor comments
Point 2: The results measured at 24 and 48 hours after start are not given in Figure 4.
Response 2: Removed 24 and 48 hours after start of recovery from figure description, since no results thereof are shown.
Point 3: Letters showing G (glycogen) and C (carboxysome) are not written or invisible in Figure 5.
Response 3: Increased font size and changed colour of descriptions in front of dark structures to white, specifically TM (thylakoid membrane), G (glycogen), and C (carvoxysome).
Point 4: It seems to me that the result of TEM picture (Figure 5 (A)) shows a mixed state of resuscitating cells at (66h -Na + KNO3). On the contrary, one ghost cell can be seen in the TEM picture given in Figure 5 (B) (right side bottom) in spite of the cell under the same situation of (66h -Na + KNO3).
Response 4: Marked ghost cells visible in panel A
Point 5: “Cyanophycn” in the subtitle “3.4. Consequential effects on cyanophyicn and the cytoplasm” should be “Cyanophycin”.
Response 5: Line 416: changed “cyanophyicn” to “Cyanophycin”
Point 6: In conclusion, periods are doubly put after [36] in the phrase “during autophagy of eucaryotic cells [36]”. Therefore, one period should be removed.
Response 6: Line 501: Deleted one of the periods after [36], also changed reference number [36] to [39] due to addition of new references
Round 2
Reviewer 2 Report
The authors have well revised the previous manuscript according to my comments. So, I would like to recommend the editor to accept the revised manuscript.